# Voices from the North: Exploring Sámi People's Perspectives on Environmental Change and Mental Well-Being: A Systematic Literature Review [†]

Valesca S. M. Venhof [1,2,*], Carolyn Stephens [3] and Pim Martens [1]

1    University College Venlo, System Earth Science, Maastricht University, 5900 AA Venlo, The Netherlands; p.martens@maastrichtuniversity.nl
2    Department of Environment and Health, GGD Groningen, 9713 GW Groningen, The Netherlands
3    UCL Bartlett Development Planning Unit, University College London, London WC1H 9EZ, UK; c.stephens@ucl.ac.uk
*    Correspondence: v.venhof@maastrichtuniversity.nl
†    The original text was submitted as an MSc Public Health graduation report/Project (year 2023) at the London School of Hygiene and Tropical Medicine.

**Abstract:** Circumpolar Indigenous People, such as the Sámi, confront significant challenges stemming from environmental shifts and interrelated issues, profoundly affecting their mental health. Nonetheless, they possess invaluable knowledge and capabilities to navigate and adapt to these transformations. This review aims to investigate peer-reviewed scientific literature, exploring the nexus between environmental changes and mental well-being within the broader Circumpolar Indigenous community, with a special focus on the Sámi People. Conducting a systematic literature review with two arms, one encompassing the broader Circumpolar Indigenous population and the other focusing specifically on the Sámi, followed by thematic analysis, we delved into their experiences of environmental changes, perceptions regarding the intertwining of environmental shifts and mental well-being, and insights into protective factors and resilience-promoting elements. By engaging with Indigenous perspectives, public health initiatives can pinpoint and leverage existing strengths within Indigenous communities and families to bolster their ability to navigate environmental shifts and safeguard mental well-being. However, our review highlighted a lack of scientific investigation of 'strength-based' factors fostering mental resilience among Indigenous populations inhabiting the Circumpolar North, such as the Sámi. Future qualitative research can address this gap, incorporating the viewpoints of individual Circumpolar Indigenous groups to explore both their distinctiveness and interconnectedness.

**Keywords:** mental well-being; environmental change; perspectives; Sámi; Circumpolar Indigenous People; resilience; protective factors; strength-based approach





## 1. Introduction

Climate change and interrelated issues such as biodiversity loss and environmental pollution are increasingly impacting mental well-being at a global level [1–3]. Direct exposure to extreme weather events such as heatwaves, flooding, and droughts is linked to mental stress, exacerbation of existing mental health conditions, and heightened rates of psychiatric admissions and suicides [4]. Additionally, climate change leads to forced displacement and directly damages healthcare facilities [5]. The indirect effects on mental health, such as increased disease and injury risks and threats to social environments, are less understood [6].

Scientific understanding of the relationship between environmental changes and mental health is evolving but remains somewhat limited [7,8]. Recent research has shed light on 'Earth emotions' [9], such as 'solastalgia' [10], 'climate anxiety' [11], and 'eco-grief' [12],

notably impacting vulnerable demographics such as youth [13] and Indigenous communities. Charlson and others [7,14,15] emphasize the urgent need for a deeper understanding of vulnerabilities and factors 'protecting and promoting resilience' in addressing climate change-related mental health challenges globally. Research regarding the psychological effects of disasters such as wildfires underscores the importance of assessing both risk levels and human resilience [16]. Strong social connections post-disaster, for example, are highlighted as 'protective' for mental well-being [17].

Indigenous populations uniquely illustrate a pronounced disparity in which vulnerability and strong resilience are distinctly evident. However, they are frequently only depicted in a 'deficit-based' manner [18–20], highlighting vulnerability, health problems, and limited healthcare access [21–25], stemming from historical injustices and colonialism [25–28]. On the other hand, resilience among Indigenous communities has been notable for millennia [29], attributable to their adaptability and deep connection to their environment [30–32]. Approximately 476 million Indigenous People worldwide [33], such as the Mayas in Guatemala and the Aborigines of Australia [34], possess unique traditional knowledge tied to their strong connection with the land [35]. They view health holistically [18,36,37], considering social, spiritual, and environmental factors crucial to well-being [26,38].

Ford et al. [29] describe 'resilience' in Indigenous People as encompassing coping, adaptive, and transformative capacities, enabling individuals to persist or transform in the face of environmental changes. They [29] stress a holistic, 'strength-based' approach to Indigenous People's mental health, focusing on 'protective factors' [26,39] while 'acknowledging vulnerability' [29]. However, there's a gap in understanding how environmental shifts affect the mental well-being of Indigenous communities [7] and an even greater gap in knowledge on 'strength-based' 'protective factors' [7,14,40] and resilience contributors [4,14].

Furthermore, existing studies on the interconnection between environmental change and mental well-being in Indigenous populations often take a quantitative approach and overlook Indigenous perspectives and traditional knowledge [41]. A growing body of collaborative research acknowledges the value of integrating Indigenous voices and perspectives [26,37,42–46]. For Indigenous Peoples, 'perspectives' encompass nuanced and dynamic ways [29,44,47] of perceiving and understanding various subjects intertwined with their beliefs, experiences, and deep-rooted connection to their land [35,48].

This systematic literature review explores the above-described gaps. Taking a 'strength-based' approach, this study aims to investigate peer-reviewed scientific literature, exploring the nexus between environmental changes and mental well-being within the broader Circumpolar Indigenous community, zooming in on the Sámi. Conducting a systematic literature review with two arms, one encompassing the broader Circumpolar Indigenous population and the other focusing specifically on the Sámi, followed by thematic analysis, we delved into their perspectives on the interconnection between environmental changes and mental well-being, zooming in on protective and resilience promoting factors rooted in their unique strengths.

In this paper, 'protective factors' and 'factors promoting resilience' are approached as distinct entities, notwithstanding their frequent interchangeable usage, both pivotal in shaping individuals' and/or communities' adeptness in confronting challenges. Protective factors reduce risk and serve as buffers against negative impacts, such as access to mental health services and community networks [14,40]. Factors promoting resilience relate to the concept of resilience itself, which varies across contexts [16]. In general, resilience involves the ability to bounce back from difficult experiences and grow stronger. Factors promoting resilience, therefore, refer to qualities, skills, or resources that enable individuals to adapt positively to adverse events or stress. While vulnerability is often seen as resilience's counterpart, both can coexist in individuals [16].

Our research underscores the distinctiveness of individual Indigenous populations, each possessing a unique history, knowledge system, and coping mechanisms [32,40,49].

Accordingly, our focus is directed toward the Arctic region, where the impacts of environmental transformations loom large [50,51]. Environmental changes exert multifaceted impacts on the mental well-being of Circumpolar Indigenous populations. They disrupt traditional livelihoods and mobility, eroding the bedrock of their existence [52]. For half of Circumpolar Indigenous peoples, such as the Sámi, reindeer herding is their primary income source [31]. Cultivated over millennia [30], this 'human–reindeer ecosystem' has historically exhibited resilience in the face of climatic shifts [31]. Nonetheless, contemporary environmental shifts exert pressures on their resilience, exacerbating pre-existing socioeconomic disparities vis-à-vis non-Indigenous counterparts. These disparities manifest in lower educational achievements, diminished life expectancies, and income differentials, alongside heightened rates of substance abuse, alcohol consumption, and suicide within Indigenous populations [40,49,52].

In our exploration of 'uniqueness,' while also recognizing the potent common core values and knowledge shared among (Circumpolar) Indigenous communities [53], we delve into the perspectives of Circumpolar Sámi people regarding protective factors and resilience-promoting elements. This inquiry is juxtaposed with an examination of perspectives on similar factors within one specific Indigenous population among the forty within the Circumpolar North, namely the Sámi [54,55]. The Sámi are of particular interest because of their status as a relatively distinct and underexplored subset within the broader Circumpolar Indigenous spectrum, exhibiting comparatively superior health outcomes [52,55]. Thus far, the underlying determinants of this disparity remain inadequately understood.

Insights into Circumpolar Indigenous Sámi perspectives and local contexts on 'strength-based factors' [19,20,29], placing these in the context of what is known about these factors for the Circumpolar Indigenous group in general, is pivotal for identifying vulnerable groups and designing culturally sensitive interventions [40,49] aimed at supporting resilience and promoting mental well-being.

*Research Objectives*

The following objectives guided the systematic literature review:

- To explore Circumpolar Indigenous People's perspectives, as described in published peer-reviewed scientific literature, on the interconnection between environmental change and mental well-being, with a special focus on their perspectives on protective factors and factors promoting resilience.
- To explore Sámi People's perspectives, as described in published peer-reviewed scientific literature, on the interconnection between environmental change and mental well-being, with a special focus on their perspectives on protective factors and factors promoting resilience.
- To compare the findings of the first and second objectives.

## 2. Materials and Methods

The systematic literature review followed the PRISMA 2020 guidelines [56]. The two arms of the systematic review had their own distinct search question, utilizing the 'PICS' (Population, Intervention, Comparator, Outcome, Study design) framework for qualitative inquiries, akin to the 'PICo' framework [57]. Both searches followed a systematic approach [58–60] conducted by the first author. The search questions were as follows:

1. What are Circumpolar Indigenous People's perspectives on the interconnection between environmental change and mental well-being, as described in published peer-reviewed scientific literature?
2. What are Sámi People's perspectives on the interconnection between environmental change and mental well-being, as described in the published peer-reviewed scientific literature?

## 2.1. Study Population: The Sámi

'Sápmi,' the homeland of the Sámi [30], spans Norway, Sweden, Finland, and the Kola Peninsula in Russia, with approximately 45,000 residing in Norway, 20,000 in Sweden, 8000 in Finland, and 2000 on the Kola Peninsula [39,55]. The Sámi People are known by various names, such as 'Sámi' in their own languages, 'Saami' as often used in the English language, and 'Lapp or Laplander or Laplandic' used in old history. The latter is nowadays seen as disrespectful. The Sámi language is a Finno–Ugric language, which can be divided into nine different dialects [55], of which some are endangered. Traditional Sámi livelihoods include handicraft, fishing, gathering, hunting, and reindeer herding [61].

Since the 20th century, the Sámi have mobilized for their rights and political voice, establishing three representative bodies—Sámi Parliaments in Norway, Sweden, and Finland—dedicated to addressing 'Sámi concerns' [30]. Engaging actively in national and global Indigenous networks such as the Sámi Council, they advocate for Sámi interests, rights, and culture [30]. Sámi communities have also undertaken legal and political measures to safeguard their land, resources, and cultural heritage.

Despite only 10% of today's Sámi population engaging in reindeer herding [61], the essence of Sámi identity and well-being remains deeply intertwined with this practice [55,62]. This interconnection, coupled with their resilience in adapting to environmental changes [39,47,63], has fostered unique traditional ecological knowledge [32,47] passed down through generations [31]. Yet, modern pressures from climate change and resource extraction industries disrupt this balance [32,49,64], impacting traditional lifestyles and stressing their resilience [26,32,39,65]. Mental health concerns, including depression and anxiety, are also a rising concern among the Sámi [49,54,66].

## 2.2. Search Strategy

For our initial systematic search (A), we specifically focused on published peer-reviewed literature reviews. This approach was chosen to gather comprehensive insights into the Circumpolar Indigenous population when approached as a cohesive entity; for the second search on Sámi People's perspectives (B), published peer-reviewed (primary and secondary) qualitative scientific research articles were included.

## 2.3. Eligibility Criteria

The eligibility criteria for both searches are included in the Supplementary File, Tables S1 and S2. For search B on the Sámi, in summary, English peer-reviewed scientific research articles published from the year 2000 and later, of which a full report was available, were included when concerned humans, concerned Sámi specifically, described perspectives and were related to the interconnection between environmental change and mental well-being.

In a scientific field in which relatively little is published on the topic, an 'inclusive approach' was taken [67]. Only literature published since 2000 was considered, aligning with the increasing attention to environmental change since Paul Crutzen introduced the concept of 'the Anthropocene' in 2000, as 'the period in which human activity started to change Earth's systems' [68].

## 2.4. Databases

For both searches, the same eight electronic databases were searched and accessed by the University of London Online Library electronic journals and databases. Six common databases and two Indigenous databases were selected on the basis of their content information, which included Medline, Global Health Database, Web of Science, PsychINFO, CINAHL Complete, Embase, Circumpolar Health Bibliographic Database, and Arctic Health database. The details on the content of these different databases are described in the Supplementary File, Table S3. As the first search on Circumpolar people's perspectives focused on reviews, the Cochrane Library Database [69] was added, resulting in nine databases for search A.

## 2.5. Search Concepts

For search A, the search concepts [58] were 'Circumpolar Indigenous People,' 'perspectives,' 'environmental change,' and 'mental health,' separated by the Boolean operator 'AND.' Synonyms for the different search concepts were added, using the Boolean operator 'OR' to combine search concepts and synonyms. Truncation and adjacency were applied where appropriate. The names of different Circumpolar populations, such as the 'Inuit,' were not separately included in search A, as this arm of the review targeted the broader Circumpolar Indigenous population. For search B, the same search concepts and synonyms were used, as illustrated in Table 1, with the difference that 'Circumpolar Indigenous People' was replaced by 'Sámi.'

**Table 1.** Search concepts and synonyms; search on the Sámi.

| Concepts | Sámi People | Perspectives | Environmental Change | Mental Health |
|---|---|---|---|---|
| Concepts with synonyms | sami OR samen OR sapmi OR sameednam OR saepmie OR sabmie saami* OR reindeer herder* OR sabme* OR scandinavian native* | perspective* OR view* OR approach OR approaches OR experience* OR belief* OR theory OR theories OR perception* OR attitude* OR idea OR ideas OR ((traditional OR ecological OR indigenous) ADJ3 (knowledge OR concept* OR framework* OR opinion* OR interpretation*)) | ((land OR soil OR environment* OR habitat OR ecosystem OR climat* OR ecological) ADJ3 (degradation OR collaps* OR change* OR shift*)) OR global warming OR greenhouse effect | (mental OR psychological OR emotional) ADJ3 (well?being OR wellness OR health) |

asterisk (*): truncation symbol.

'MeSH' subject headings were applied to each key search concept [70]. After refining the search strategy, it was discussed via email with the LSHTM librarian. The final strategies were first implemented on the MEDLINE database and then tailored for each individual database. Due to technical challenges, hand-searching was applied to the Arctic Health database and the Circumpolar Health Bibliographic Database.

Furthermore, 'snowball sampling' was employed, involving the screening of reference lists from included articles [71]. Detailed search strategies for the various databases are provided in the Supplementary File, Tables S4–S20. References, including article content, abstracts, and keywords, were imported into EndNote 20 reference management software from each database. Duplicate entries were removed using EndNote 20's de-duplication function.

## 2.6. Study Selection

The stages of study selection, quality assessment, and data analysis were conducted consistently across both searches. When meeting the eligibility criteria (Section 2.3), the reference, with full-text pdf attachment, was copied to a separate folder in Endnote. Then, the full-text article of every reference in that folder was screened, applying the same eligibility criteria. Articles not meeting the eligibility criteria were removed from the folder.

For the remaining articles, a data summary chart in Microsoft Excel version 16.85 was used to gather information on publication characteristics (title of study, author, source, and year of publication), population characteristics (country and population of interest), interest, context, study characteristics (type of study, sample size, and method of data collection), and a short summary of findings.

## 2.7. Quality Assessment

The selected articles were critically appraised [58], applying the ten items 'CASP checklist for systematic reviews, and for qualitative studies' [72]. An item scored '0.5 points'

(instead of '0 or 1' [72]) when the answer to that item was considered 'partly true.' All articles were scored on a ten-point scale, with 'eight to ten' defined as 'high quality (green),' 'five to seven' as 'moderate quality (orange),' and 'one to four' as 'weak evidence (yellow)' articles.

Furthermore, for each article, the 17 items described in the 'CONSIDER statement checklist for reporting research involving Indigenous People' [73] were scored. Table S22 in the Supplementary File includes an original copy of this checklist.

### 2.8. Data Analysis

We adhered to Aveyard's [58] guidelines for conducting thematic analysis and implemented the 'thematic synthesis' method outlined by Thomas and Harden [67]. The latter [67] delineates three stages for thematic analysis, commencing with 'line by line' screening and coding. Each included article's full text, encompassing abstract and Section 3, was transferred to MS Word. There, all text describing perspectives or experiences was highlighted. The highlighted segments were then compiled into an MS Excel file, where individual text parts were summarized and translated into concepts, or 'themes' [58,67], maintaining proximity to the original text.

In the subsequent stage [67], themes in each category underwent comparison to identify commonalities and distinctions. Similar themes were collated [58], retaining the visibility of the original text and its source. New 'overarching' themes [67] were developed, culminating in a 'hierarchical tree structure of descriptive themes' [67]. A summary of findings was crafted for each descriptive theme [67].

The final stage centered on 'interpretation' [67,74] of the 'descriptive themes,' yielding several principal 'analytical themes' [67]. The distinction between 'descriptive themes' from the second stage and 'analytical themes' from the third lies in the former's adherence to the original text, while the latter transcends primary studies to generate new interpretive constructs, explanations, or hypotheses [67]. Themes prevalent in robust articles [67] were characterized by high-quality assessment scores as outlined in Section 2.7, and those recurrent across multiple articles were accorded greater 'weight' or significance [58].

For the third objective of our study, we conducted a comparative analysis of the themes identified in both searches using a separate Microsoft Excel spreadsheet. This analysis revealed both similarities and differences between the overall Circumpolar Indigenous population and the Sámi specifically.

### 2.9. Ethical Considerations

This systematic literature review was conducted as a graduation (year 2022/2023) project of the first author for her master's in Public Health (environment and health) at the London School of Hygiene and Tropical Medicine (LSHTM). Approval from the LSHTM medical ethics committee was sought but deemed unnecessary for this research. Given the involvement of Indigenous People, the authors, though not Indigenous, tried to approach the research process and interpretation of findings in a discrete and respectful manner, keeping as close to the original text as possible. Recognizing the importance of collaboration with Indigenous scholars, as also recommended by the CONSIDER statement of research involving Indigenous People [73], the involvement of an Indigenous researcher would have been of great value to ensure accurate interpretation of themes and concepts from the Indigenous perspective rather than a Western one. However, due to the requirement for this project to be completely independently conducted by the first author for her master's project, no Indigenous collaborators were involved.

## 3. Results

### 3.1. Summary of Results

The initial search (A) on perspectives of the overall Circumpolar population resulted in 47 articles. After screening the title, keywords, and abstract of these articles on eligibility criteria, five articles remained. One additional article was added by hand-searching,

resulting in a total of six articles included for full-text screening. After full-text screening, only two systematic reviews remained for quality assessment and thematic synthesis.

The second search (B) on Sámi People's perspectives identified 53 articles. After deduplication, 40 articles were screened by title, abstract, and keywords, of which 11 articles were eventually included for full-text screening. Another nine articles were added after hand-searching the Arctic database and reference searching (see Section 2.5). After a full-text screening of 20 articles, eight articles remained for further quality assessment and thematic synthesis.

There was a concern regarding potential overlap in data sourced from articles included in both searches. However, this concern appeared largely unfounded, with the exception of Furberg et al.'s article [75], which featured in both Lebel et al.'s review [49] on Circumpolar People (search A) as in the Sámi search (search B) as a high-quality reference (search B). We exercised caution to prevent misinterpretation or duplication of findings.

Furthermore, Macdonald et al.'s review [40] included in search A encompassed findings from three Sámi-related articles, all authored by Bals et al. [40]. Despite these articles not surfacing in the Sámi search (B) results, we meticulously screened their full texts. It became evident that these articles collectively depicted findings from a single study unrelated to environmental change and that the review's broader scope largely outweighed the influence of these articles. Once again, we were vigilant in avoiding redundant interpretations and presentations of data.

Figures S1 and S2 in the Supplementary File display the PRISMA flow diagrams [56] for both searches. While traditionally, one systematic literature review necessitates a single PRISMA diagram, for enhanced clarity, the findings are presented in two distinct diagrams.

### 3.2. Study Characteristics

Search A comprised two systematic reviews [40,49], with one focusing specifically on climate change [49] while the other explored the broader spectrum of social and environmental changes [40]. The latter, authored by MacDonald et al. [40], delved into the experiences of Indigenous Circumpolar youth, contrasting with Lebel et al.'s [49] comprehensive study involving Circumpolar Indigenous Peoples across all age groups. Notably, Lebel et al. [49] exhibited a pronounced overrepresentation of the Inuit, featured in 16 out of 26 included articles. Conversely, MacDonald et al.'s [40] review primarily encompassed four Circumpolar Indigenous populations, with a notable emphasis on Alaska, where 8 out of 15 reviewed articles pertained to Indigenous People from the region, including Alaska Natives and Inupiat.

Search B centered on the Sámi community, incorporating eight articles, comprising one systematic review [55] and seven qualitative studies [41,62,75–79]. Among the qualitative studies, five were interviews; one employed a qualitative mixed-method design [79], and another integrated both qualitative and quantitative methodologies [78], utilizing meetings, discussions, and dialogues. Notably, all qualitative studies featured relatively modest sample sizes, ranging from $N = 9$ [62] to $N = 60$ [78], with four studies specifically focusing on young Sámi [41,62,76,78]. While three of the eight studies, including Jaakkola et al.'s systematic review, centered on the impact of climate change [55,62,75], the remaining studies provided insights into broader socioeconomic, cultural, and environmental shifts.

Within search B, studies pertaining to Swedish Sámi [75,76,78] comprised three out of eight articles, with one specifically addressing Sámi residing in the Swedish–Finnish border region [77], two focusing on Norwegian Sámi [41,62], and one detailing the experiences of Sámi in Finland [79]. Notably, no articles elucidating the perspectives of Russian Sámi were uncovered. While Jaakkola et al.'s systematic review [55] did include a small fraction of articles discussing Russian Sámi, constituting '4% of the 294 analyzed reports', the bulk of the literature centered on Norwegian Sámi [55].

Figures S3 and S4 in the Supplementary File show the detailed study characteristics of the articles included in both searches A and B.

### 3.3. Quality Assessment

For search A, both included articles were of moderate to good quality. For the second search (B) on the Sámi, the articles of Tervo and Nikkonen [79] and Kaiser et al. [76] scored highest, followed by the articles of Furberg et al. [75] and Nystad et al. [41]. More details on the results of the quality assessment can be found in the Supplementary File, Table S21.

The 'CONSIDER checklist' [73] was used to report health research involving Indigenous Peoples. All articles in searches A and B focused on Indigenous Peoples, yet they lacked significant details on their involvement. The two articles in search A [40,49], both systematic reviews, failed to provide any specifics regarding reporting on Indigenous Peoples.

For search B, the study by Tervo and Nikkonen [79] stands out for its comprehensive coverage of considerations pertaining to reporting on Indigenous Peoples, addressing six out of 17 criteria. Following closely, the research by Nystad et al. [41] touched upon four out of 17 criteria. Conversely, one study failed to provide any detailed insights [77], while another only covered two criteria [78]. The remaining four articles merely addressed one criterion each. Table S23 in the Supplementary File outlines the detailed findings of the 'CONSIDER appraisal' for the top two scoring articles. Among the articles from both search A and B, 'relationships' emerged as the most frequently discussed 'CONSIDER item.' This encompasses aspects such as the team's expertise in Indigenous health and research, as well as 'prioritization,' delineating the process of formulating research objectives (73).

### 3.4. Thematic Analysis

In accordance with Section 2.7 addressing quality assessment, emphasis was placed on high-quality articles. In search B, for instance, priority was assigned to the four exemplary articles [41,75,76,79]—see Supplementary File, Table S21, for specifics—over the two articles garnering a lower score of five points [62,77]. Comparison of themes found in search A and B resulted in three overlapping primary themes: encounters with environmental changes (Section 3.4.1), perceptions of the interrelation between environmental change and mental well-being (Section 3.4.2), and perspectives on 'strength-based factors' (Section 3.4.3). The latter, following the main focus of our study, was bifurcated into 'protective factors' and 'factors promoting resilience.' For each primary theme, we begin by presenting the results for the Circumpolar group, followed by the results for the Sámi. The latter is contextualized within the broader findings for the Circumpolar group as a whole.

#### 3.4.1. Encounters with Environmental Changes

As their very livelihoods, identities, and cultures are intricately woven with the natural environment, Circumpolar Indigenous communities have long borne witness to the consequences of climate change. Decades ago, they began to discern the signs: 'shifting temperatures, alterations in animal behaviors and habitats, dwindling land and sea ice, thawing of permafrost, shifts in vegetation and the water cycle' [49].

Similar narratives emerge from Sámi reindeer herders in Sweden and the Finnish-Swedish border region [77], describing changes such as erratic weather patterns, heightened occurrences of extreme weather events, and shifts in the rhythm of seasons [75–77]. Notably, Swedish Sámi herders report a decline in the tree line, leading to fewer trees and more bushes [75]. Consequently, tree-hanging lichen—a vital supplement to reindeer pasturage—is depleting, leading to dwindling food supplies for the reindeer and the necessity for costlier supplementary feeding [75]. Compounding these environmental shifts are anthropogenic interventions—roads, urbanization, wind farms, hydropower installations, and the burgeoning impacts of tourism—all of which constrict available grazing lands and disrupt traditional migration routes for Sámi reindeer herds [75].

Concomitant with these transformations, Sámi herders find themselves ensnared in a 'web of socio-economic pressures' [55,75,76,78]. Financial strains force them to maintain larger herds, necessitating the adoption of modern technologies such as snowmobiles and airplanes, as well as increased reliance on supplementary feeding—each adding to the financial burden [75,77]. This intensified grazing pressure, coupled with diminishing grazing lands,

perpetuates a cycle of ecological degradation and financial strain, compelling herders to further increase herd sizes, thereby exacerbating the unsustainable trajectory [75,77]. In essence, these intertwined challenges have ensnared Sámi reindeer herders in a 'vicious cycle of unsustainability' [75,77], where ecological shifts amplify economic pressures and vice versa.

### 3.4.2. Interrelation between Environmental Change and Mental Well-Being

For Circumpolar Indigenous People, environmental changes assail cultural foundations, identity, and food security, amplifying vulnerabilities [49]. Lebel et al. [49] describe various emotional reactions among Circumpolar Indigenous Peoples in response to climate shifts, ranging from interpersonal strife and internalized anguish to profound sensations of powerlessness and indignation. In regions such as Nunatsiavut and Nunavut, these reactions manifest as a complex tapestry of emotions encompassing grief, melancholy, ennui, and even contemplations of self-harm. The toll extends to the confinement of homes for prolonged durations, breeding sentiments of ennui, seclusion, and entrapment among Nunatsiavut Indigenous People [49]. These sentiments intertwine with adverse outcomes such as substance misuse, domestic violence, and suicidal tendencies [49].

In comparing findings on the interconnection of environmental change and mental well-being for the Circumpolar group and the Sámi, we identified many overlapping results. These findings are categorized into four subthemes, detailed below: 'layered burden on mental well-being,' 'worries and stress,' 'feeling ignored and threatened,' and 'community norms and the impact on identity.'

### Layered Burden on Mental Well-Being

In the findings of search A [40,49], the intricate nexus between environmental shifts and the mental well-being of Circumpolar Indigenous Peoples is underscored as inseparable from the weighty historical backdrop of 'colonization, assimilation efforts, resource exploitation, environmental degradation, and influx from southern territories' [49]. Lebel et al. [49] further detail the challenges of Circumpolar Indigenous Peoples, highlighting their 'marginalization, socio-political disenfranchisement, and the erosion of traditional livelihoods and land rights.' The psychological consequences of environmental changes appear to compound atop an entangled history of past traumas [49].

The insights from four Sámi articles in search B [55,75–77] underline the above-described 'layered burden' on mental well-being, in which Swedish Sámi reindeer herders [75,77] perceive climate change not as a standalone menace to their way of life, but rather as 'yet another burden on an already stressed industry and cultural heritage' [75].

### Worries and Mental Stress

Lebel et al. [49] delineate the profound worries, concerns, and mental stress endured by Circumpolar Indigenous communities. These encompass apprehensions regarding the potential erosion of their livelihoods, traditional practices, autonomy, and the invaluable reliance on ancestral wisdom. Additionally, they express anxieties concerning their capacity to adapt to environmental shifts, fearing a loss of cultural identity and the inability to transmit knowledge to succeeding generations [49].

Similar apprehensions resonate within the Sámi community [55,75,76], exemplified by Swedish reindeer herders grappling with the worrisome possibility of losing their ancestral knowledge, fostering a sense of 'insecurity' [76]. Sámi voices echo worries concerning the viability of reindeer herding [62,75], the preservation of traditional wisdom [75], and the endurance of their cultural heritage [55,62] into the future. Notably, the Inuit of Nunatsiavut share parallel concerns regarding the fate of traditional caribou hunting [49].

Half of the Sámi articles [55,62,75,76] described experiences of mental stress, including the pressures inherent in their traditional reindeer herding livelihoods [62,75], marked by relentless workload and a continuous sense of 'no rest' [75]. They struggled with a profound loss of equilibrium with their reindeer [75], heightened anxieties over predatory threats [75], and the weight of financial dependence [75,76]. Additionally, Sámi interviewees

articulated stress due to adaptation and mitigation measures [55] alongside a surge in conflicts [76]. Similar narratives of escalating social tensions are echoed among other Circumpolar Indigenous communities facing climate change [49].

Moreover, the Sámi expressed a profound sense of vulnerability [55,75,77], confronting existential threats to their identity and cultural heritage [75,76], with some articulating 'grief for the future' [62,75,77]. In Furberg et al.'s study [75], young Sámi reindeer herders summarized their situation as reaching the point in which they can no longer adapt, 'facing the limits of resilience' [75]. Such feelings resonated with Swedish reindeer herders as well, as highlighted in the works of Heikkinen et al. [77] and Kaiser et al. [76].

Feeling Ignored and Threatened

Circumpolar Indigenous communities in Alaska felt a 'lack of institutional support' [49], compounding their challenges.

Similarly, Swedish Sámi described the profound psychological toll of environmental shifts, exacerbated by a sense of 'being ignored' and controlled by society and institutions [75–78]. They expressed a substantial sense of powerlessness, lamenting their exclusion from decision-making processes [75] and grappling with issues they experienced as stemming 'from the actions of others' in the context of climate change [75]. Moreover, Swedish Sámi voiced feelings of marginalization and disrespect within society [75,78], highlighting the 'unequal playing field' dominated by industries where the 'reindeer industry must always take a back seat' [75].

Community Norms and the Impact on Identity

For Circumpolar Indigenous People, environmental change profoundly impacts their identity [49].

This holds particularly true for the Sámi, where reindeer herding holds esteemed status within their culture, revered as 'a way of life' and a source of pride [78]. Environmental shifts encroach upon cherished facets of Sámi identity, such as 'independence,' 'hardiness,' and 'autonomy' [41,62,78,79]. In the research conducted by Kaiser et al., young Swedish reindeer herders articulated the notion of 'having a culture of their own,' embodying a 'macho culture' [76] characterized by community norms emphasizing perseverance and self-reliance, wherein abandoning reindeer herding amid challenging circumstances is viewed as 'failure' and a 'forfeiture of identity' [76]. Moreover, Sámi individuals often feel compelled to persevere and 'just endure' [41,75,76] with no alternative. These community norms exert significant mental strain on young Sámi, fostering a culture where open discussion of problems is uncommon [75]. On the flip side, Swedish Sámi male reindeer herders perceive the family as a vital sanctuary for receiving support during psychological challenges [76,78].

Regarding their transition into reindeer herding, young Swedish Sámi reindeer herders grapple with a 'paradox of being free yet bound' [76]. While they have the freedom to choose this path, there is a profound sense of pride and communal expectations, such as the imperative to be resilient, which intensifies the pressure to 'uphold the family legacy of reindeer herders' [76]. Echoing this sentiment, Omma et al. [78] observed that many young Swedish male Sámi participants view their culture as a holistic lifestyle, encompassing distinctive ways of 'experiencing, thinking, and learning' [78], with an overwhelming urge to 'preserve and fortify their cultural heritage and language' [78].

3.4.3. Protective Factors

Figure 1 provides a visual summary of the protective factors found for the Sámi, presented clockwise, with 'strong family ties' being the most strongly supported, found in six articles, followed by 'ethnic identity and pride,' described in four articles, succeeding with 'participation in community,' 'reindeer herding,' 'Sámi language,' traditional practices,' 'interconnection with the environment,' 'traditional knowledge,' and 'physical

activity.' Further details on the weighting of factors can be found in the Supplementary File, Table S24.

**Figure 1.** Protective factors, as found for the Sámi (search B). Factors are clockwise ordered, with the strongest evidence for 'strong family ties,' followed by 'ethnic identity and pride.'.

Comparing the protective factors found for the broader Circumpolar population and the Sámi, we detected that these factors largely overlapped except for 'physical activity,' which, although weakly supported, was noted specifically as a protective factor for the Sámi. Additionally, while there are many other Circumpolar reindeer herders, such as the Nenets, or caribou herders, such as the Inuit, 'reindeer herding' was identified only as a protective factor for the Sámi. The various protective factors are described in more detail below. Again, we first present the results for the Circumpolar group, followed by the results for the Sámi.

Strong Family Ties and Community Participation

In both reviews of Circumpolar Indigenous populations, a resounding emphasis was placed on the pivotal role of robust familial and communal ties in safeguarding mental well-being [40,49]. For instance, within Inupiat communities, the significance of interpersonal and communal bonds, intertwined with 'cultural integrity,' emerged as vital for navigating environmental changes and fostering successful adaptation [49]. MacDonald et al. further elucidated protective factors at both the community and family levels [40]. These encompassed the 'presence of positive role models, secure environments, a sense of belonging, and active, meaningful engagement' at the community level, while 'close parental relationships, safety, feeling valued, and strong kinship structures' were highlighted at the family level [40].

Similar findings resonated among the Sámi population, where 'strong family ties' [41,55,62,76,78,79] and 'active participation within the Sámi community' [41,55,78,79] were recurrently underscored as paramount for mental well-being. Sámi adolescents in Norway underscored the significance of 'cultural continuity' [41], evidenced by their involvement in 'traditional ceremonies and strong social networks with extended family and godparents' [41]. Swedish male Sámi reindeer herders emphasized that 'the family is a kind of system of service you can rely on when you need practical help, and you are supposed to be there, show loyalty and stick together' [76].

Language

MacDonald et al. recognized the importance of native language retention among Circumpolar Indigenous youth and its positive impact on mental well-being [49].

Similarly, the preservation and use of the Sámi language emerged as a significant protective factor across half of the included Sámi studies [41,55,78,79]. For young Sámi individuals in Sweden and Norway, fluency in the Sámi language was deemed essential for complete integration into the Sámi community [41,78].

Interconnection with the Environment and Traditional Knowledge

For Indigenous peoples inhabiting the Circumpolar North, their physical and mental well-being, as well as their culture and identity, are intricately woven with the natural environment and the traditional practices they engage in on the land and at sea [49]. Lebel et al. emphasize this profound interconnection as a pivotal factor fostering resilience among Circumpolar Indigenous communities, constituting a 'core component of Inuit, Inupiat, and Gwich'in identities' [49]. Similarly, this bond with the environment emerges as a cornerstone of identity and culture among the Sámi, as observed in studies by various researchers [41,78,79]. Furthermore, the same sentiment was expressed by Circumpolar Indigenous youth regarding traditional knowledge [49].

The Finnish Sámi articulate that the natural environment is 'much more than a source of livelihood or an object of interest; it is home, lifestyle, ethnic history and future' [79]. Norwegian Sámi adolescents express that being immersed in nature alleviates stress, offering them a sense of liberation and an 'opportunity for introspection' [41]. Furthermore, for young Swedish Sámi, engagement in traditional outdoor activities such as 'fishing, hunting, reindeer herding, and berry picking' serves to fortify community resilience [78]. Additionally, active involvement in preserving traditional (ecological) knowledge emerged as another important protective factor [41,78],

Traditional Practices and Cultural Identity

For Circumpolar Inuit and Gwich'in communities, engagement in traditional sea ice and land-based activities is deeply intertwined with feelings of 'cultural pride, self-value, confidence, motivation, and sense of self-determination' [49]. Similarly, among Nunatsiavut youth, involvement in land-based activities not only cultivates 'supportive social networks and positive role models' but also 'nurtures trust and solidarity' [49]. Moreover, MacDonald et al. underscore the significance of 'ethnic pride and identity' as a pivotal protective factor for Circumpolar Indigenous youth operating at the individual level [49].

Across the Sámi community, participation in traditional practices [41,55,78,79] is widely recognized as a protective factor for mental well-being. Norwegian Sámi adolescents perceive these activities as integral to fostering a 'strong and positive Sámi identity' [41], directly correlating with enhanced mental well-being. Conversely, environmental changes pose a significant threat to the strong sense of ethnic identity and pride among Sámi individuals in Sweden [75,76,78].

Involvement in Reindeer Herding and Physical Activity

The protective role of the connection with the reindeer and being physically active was only described for the Sámi. Participation in reindeer herding activities was considered a strong protective factor for their mental well-being and interwoven with their culture and identity [41,55,78,79]. Among young Sámi in Norway and Sweden, the 'siida,' or reindeer herd, plays a pivotal role in fostering social cohesion, connection to the land, and interpersonal support networks [41,76,78]. It serves as a locus for not only tending to the reindeer but also for meaningful social interaction with peers and extended family, providing both psychological solace and practical assistance [41,76,78].

Moreover, Finnish Sámi underscored the vital role of 'work-related physical activity' in promoting well-being, emphasizing its foundational importance [79].

### 3.4.4. Factors Promoting Resilience

MacDonald et al. [40] delineated 17 protective factors crucial for bolstering the mental well-being of Circumpolar Indigenous youth, operating primarily at the individual level. Many of these factors are intricately intertwined with familial and communal dynamics, as previously outlined, including facets such as 'ethnic pride' and 'physical presence within one's home community' [40]. However, several factors delve into individual-based skills and coping mechanisms, aligning with our conceptualization of 'factors promoting resilience' as expounded in our Section 2. These 'factors promoting resilience' encompass the facets of 'belief in self,' 'sense of purpose,' 'responsibility towards oneself, family, and/or community,' and 'self-reliance' [40].

Thematic analysis unveiled a spectrum of resilience-promoting factors perceived by the Sámi community. These are summarized in Figure 2. The Sámi appear to place a high value on 'independence' and 'autonomy' [41,62,78,79]. Within half of the Sámi literature examined [75–78], 'individual coping mechanisms' [76] were elucidated, often ingrained from childhood, such as 'taking ownership of individual actions' [75,78,79], maintaining resilience and optimism, or 'carry on and be positive' [75,76,79], and cultivating 'self-trust, confidence, and perseverance' [41,78]. In navigating challenging circumstances, Kowalczewski and Klein [62] found that their participants predominantly described individual coping mechanisms, such as 'journaling, solitary retreats to reindeer pastures, and hiking,' rather than communal strategies such as 'seeking support from friends and family or engaging in group activities' [62].

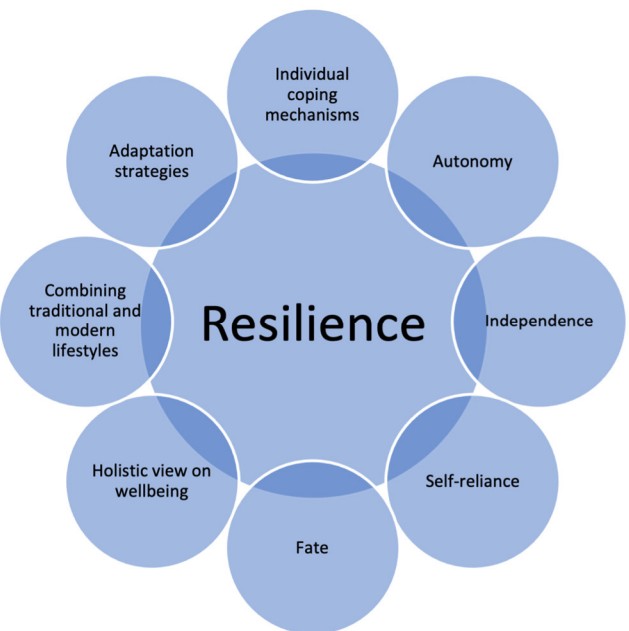

**Figure 2.** Factors promoting resilience, as found for the Sámi (search B).

Omma et al. assert that fostering 'self-reliance,' particularly 'in terms of acquiring knowledge and skills,' is instrumental in cultivating and preserving a 'sense of self-worth and ethnic pride' among Sámi youth [78]. Furthermore, Tervo et al. underscore the significance of acknowledging 'fate' as a force 'beyond a specific deity' [79], alongside advocating for a 'holistic perspective on well-being' [79]. Additionally, findings indicate that both Sámi adolescents in North Norway and young Sámi individuals in Sweden value the capacity to integrate traditional lifestyles with 'modern Western' influences as a means to alleviate mental strain [41,78].

#### Adaptation Strategies and Active Engagement

Amidst the enormous challenges faced by Indigenous Peoples inhabiting the Circumpolar North, Lebel et al. underscore the presence of 'optimism, resilience, and innovative

adaptation strategies' [49]. MacDonald et al. shed light on the proactive stance of young Circumpolar Indigenous individuals, who assertively 'advocate for themselves,' leveraging available resources and 'seeking guidance to navigate through adversities' [40]. For Circumpolar communities such as those in Alaska, active participation in 'Indigenous-led initiatives,' particularly in 'climate action and monitoring,' serves not only to alleviate mental strain but also fosters a sense of 'empowerment, resilience, and connection to their ancestral lands, traditional diet, and cultural heritage' [49]. Here, access to financial resources and, thereby, technical ways to adapt were mentioned as important prerequisites for strengthening adaptive capacity [49].

Similarly, Swedish Sámi reindeer herders expressed feelings of hope and perseverance for the future, alongside a determination to discover novel methods of adaptation [75,76]. In the study conducted by Furberg et al. [75], Swedish Sámi reindeer herders exhibited a profound sense of 'personal accountability for their sustenance and survival, demonstrating a commitment to eco-friendly practices such as reducing motorized transportation, eschewing helicopter herding, and responsibly managing waste' [75]. However, some acknowledged a struggle between the desire to resist challenges and a depletion of energy to confront them (75). Although Sámi actively engaged and advocated for their rights and interests [30], their activism was not found in the articles as a resilience-promoting factor.

*3.5. Summary of Findings*

Figure 3 encapsulates the 'strength-based factors' for fostering mental well-being in the context of environmental change within the Sámi community. The inner part of this 'flower' illustrates the factors promoting resilience, while the outer 'leaves' represent the protective factors. Due to the weak strength of evidence for the protective factor 'physical activity' and its apparent connection to reindeer herding, this factor is depicted as a small leaf attached to the reindeer factor. Resonating the findings presented in the Section 3, we emphasize the importance of a supportive environment, highlighting the necessity of access to financial resources, institutional and societal support, and the recognition of historical context.

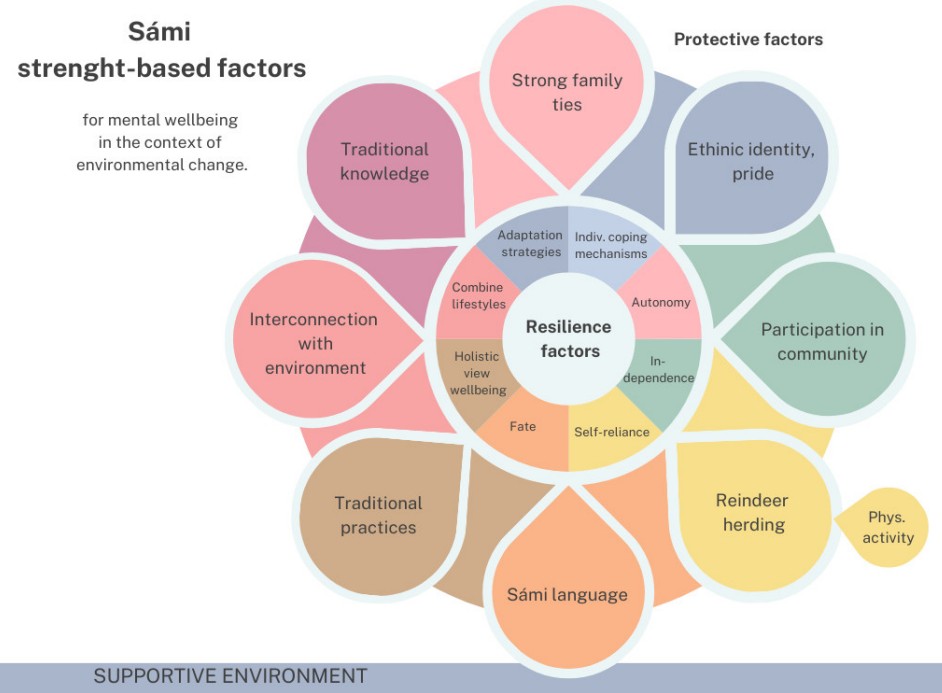

**Figure 3.** Summary of factors contributing to Sámi People's mental well-being in the context of environmental change. This figure is created from a template from www.canva.com, accessed on 14 September 2023.

Figure 3 should be interpreted as exploratory, a non-rigid scheme that can guide and inspire future research.

## 4. Discussion

Our review shows that Circumpolar Indigenous communities are experiencing profound psychological challenges related to climate change and anthropogenic interventions, impacting their traditional livelihoods, cultural foundations, and food security. Emotional responses among these communities range from interpersonal strife to feelings of powerlessness and indignation, with reports of substance misuse and domestic violence. These emotions resonated with the Sámi reindeer herders, for whom environmental changes lead to dwindling resources, financial strain, and unsustainable practices. They experience environmental stress and mental health challenges related to environmental change, with concerns about the erosion of their cultural identity and heritage. The burden on the mental well-being of Circumpolar Indigenous People such as the Sámi is compounded by historical traumas, societal marginalization, and a sense of vulnerability [80,81], with community norms exerting pressure to persevere amidst changing circumstances.

In addition to these challenges, we identified some important 'strength-based factors' involved in protecting and promoting mental well-being among both the broader Circumpolar population and the Sámi community. Both searches emphasized the significance of strong family and community bonds, cultural or ethnic pride, and adherence to traditional practices and knowledge. While the pool of eligible articles was limited, both arms of the review yielded interesting insights warranting further investigation.

First, Tervo and Nikkonen [79] underscore the significance of a holistic approach to mental well-being. While not explicitly delineated as a distinct resilience-promoting factor in the other included Sámi articles, the collective experiences described for the Sámi consistently affirmed this holistic paradigm, wherein individual well-being is intertwined with their cultural identity, spiritual beliefs, physical health, and even the welfare of their reindeer [75,77]. This holistic perspective on well-being resonates across various Indigenous groups [29,47,82], often intertwined with their profound connection to the 'land' [35,48].

Second, it appeared challenging to categorize findings into themes, as there was an intricate interplay among diverse protective factors and resilience-promoting elements observed within Circumpolar (Sámi) Indigenous communities. For instance, among the Sámi, engagement in reindeer herding not only bolsters individual 'self-worth and ethnic pride' [78] but also fosters social cohesion through communal learning of traditional herding practices, thereby enhancing community and individual resilience and well-being. Moreover, this activity intersects with traditional ecological knowledge (TEK) and emphasizes 'physical activity.' MacDonald et al. similarly highlight this interconnectedness across factors acting at 'individual, familial, and communal levels' [40].

Third, although many of the 'strength-based factors' identified within the Sámi community resonated across broader Circumpolar Indigenous populations, there were also some notable distinctions. The Sámi seem to place a premium on 'independence and autonomy.' Numerous Sámi sources portray a narrative of resilience and fortitude, wherein the prevailing ethos entails a steadfast resolve to 'endure' [41,75,76] and confront challenges autonomously. Further examination is warranted to elucidate the role of these communal norms, individual coping strategies, and the preservation of the Sámi language. The recent United Nations report on 'Indigenous determinants of health' similarly underscores the significance of 'language' as a pivotal protective factor for the mental well-being of Indigenous Peoples [26].

The regional disparities observed within the Sámi community, particularly concerning communal norms and coping mechanisms, further underscore the 'uniqueness' of individual Circumpolar Indigenous populations and highlight the importance of local interpretations and solutions [32]. MacDonald et al. [40] further assert the absence of homogeneity among Indigenous communities across the Circumpolar North regarding 'communication and cultural norms' in addressing mental health concerns. They cite the

example of the Canadian Inuit, who tend to internalize mental health issues, contrasting with Alaska Native Inupiaq, who openly discuss such challenges. Enhanced understanding of the mechanisms through which communal norms and coping mechanisms impact mental well-being and their potential divergence from other Circumpolar Indigenous populations can facilitate tailored interventions aimed at bolstering mental resilience and adaptive capacity.

Ford et al. [29] underscore the vital significance of 'collective action,' besides 'traditional knowledge' and 'attachment to place,' in supporting mental resilience among Circumpolar Indigenous populations. Our findings highlight the necessity for further exploration into the role of active engagement in promoting mental well-being among the Sámi community. Swedish reindeer herders have expressed a profound sense of the 'urge to fight,' yet they often find themselves 'lacking the necessary energy,' influenced by a multitude of interconnected historical factors [75]. This underscores the imperative for additional research aimed at understanding their needs and identifying avenues through which they can be empowered.

### 4.1. Limitations

This study presents several significant limitations. First, the scope for generalizing findings is constrained, as only two reviews were included in search A and eight articles in search B. Within the 'search A' reviews focused on Circumpolar Indigenous Peoples, there's a notable imbalance. Lebel et al.'s review [49] predominantly features Inuit perspectives, while MacDonald et al.'s [40] emphasizes Alaska Natives and Inupiat, with a narrower focus on climate change in the former and a specific focus on youth in the latter. Furthermore, we only relied on the systematic reviews and not on the original research articles examined in these reviews.

Moreover, the studies in search B on Sámi perspectives suffer from small sample sizes, and half of them exclusively examine youth [41,62,76,78]. Additionally, the representation of Sápmi regions is uneven, with a bias towards Swedish Sámi in half of the studies [75–78].

Possibly due to the inclusive approach, four articles dominated in describing protective factors and factors promoting resilience for the Sámi [41,55,78,79], and some articles were included with limited findings on the 'strength-based factors.' However, these 'deficit' articles offered valuable insights into the Sámi's perceptions on the interconnection of environmental change and mental well-being.

Acknowledging the complexity of Indigenous knowledge, the authors recognize the risk of decontextualizing it through thematic synthesis [67]. Adhering closely to the original text, albeit due to the first author's lack of Indigenous background, might have constrained the interpretation and description of analytical themes. Furthermore, we acknowledge the absence of an Indigenous scholar's involvement in this research as a significant limitation, which might have resulted in incomplete or skewed interpretations of findings.

Furthermore, restricting the review to English-language sources and excluding grey literature potentially overlooks crucial perspectives from Circumpolar Indigenous communities, including the Sámi. Failure to engage Indigenous researchers may have further hindered a comprehensive understanding.

### 4.2. Conclusions

Focusing on perspectives, this systematic review predominantly researched qualitative data and reviews, including qualitative studies. Our findings show the invaluable significance, albeit constrained accessibility, of qualitative research integrating the perspectives of Indigenous Peoples [41]. The current body of research on the interplay between environmental shifts and the mental well-being of Circumpolar Indigenous Peoples predominantly adopts a 'deficit' perspective, focusing on risks and vulnerabilities. However, scant attention has been paid to exploring the protective and resilience-enhancing factors rooted in Indigenous strengths, particularly among populations such as the Sámi in the Circumpolar North [29,83,84].

Circumpolar Indigenous communities such as the Sámi possess rich traditional knowledge and cultural practices deeply intertwined with their environment and identity. Areas warranting deeper qualitative exploration encompass the dynamics of 'strength-based factors' interplay [40], the influence of community norms, individual coping strategies, and language for the Sámi community, as well as the distinctive variations and parallels among Circumpolar Indigenous peoples' encounters and viewpoints regarding mental well-being amidst environmental shifts. In forthcoming mental health research and policy formulation, collaboration and inclusion of Indigenous communities [75,77] stand as imperative steps to deepen comprehension of local adversities, vulnerabilities, and strengths alongside the intricate interplay of social, economic, and historical elements impacting mental well-being, as well as the (local) potential for adaptation [49].

By studying Indigenous people's perspectives, public health efforts can identify and build on existing strengths within Indigenous communities and families to support their capacity to adapt to environmental changes and to promote and protect mental well-being [49]. Integrating their perspectives can help tailor public health interventions to be culturally relevant, place-specific, and respectful of Indigenous values and knowledge. Moreover, this approach facilitates the identification of vulnerable groups [49,85] and addresses disparities in mental health outcomes across the Circumpolar North. Circumpolar Indigenous communities such as the Sámi possess a distinctive holistic perspective on health, embracing mental, physical, and spiritual dimensions. By integrating these viewpoints into research and policymaking processes, involving Indigenous voices can yield more robust public health strategies, effectively addressing mental health within a broader framework.

**Supplementary Materials:** The following supporting information can be downloaded at: https://www.mdpi.com/article/10.3390/challe15020030/s1, Table S1: Eligibility criteria, search A on Circumpolar People; Table S2: Eligibility criteria, search B on Sámi People; Table S3: Databases for electronic searching, search A and B; Table S4: Results Medline, search A; Table S5: Results Global Health, search A; Table S6: Results Web of Science, search A; Table S7: results PsychINFO, search A; Table S8: Results CINAHL Complete, search A; Table S9: Results Embase, search A; Table S10: Results Cochrane Library, search A; Table S11: Results Circumpolar Health Bibliographic Database, search A; Table S12: Results Arctic Health Database search A; Table S13: Results Medline, search B; Table S14: Results Global Health, search B; table S15: Results Web of Science, search B; Table S16: results PsychINFO, search B; Table S17: Results CINAHL Complete, search B; Table S18: Results Embase, search B; Table S19: Results Circumpolar Health Bibliographic Database, search B; Table S20: Results Arctic Health Database, search B; Figure S1: PRISMA flow diagram [63], systematic search A; Figure S2: PRISMA flow diagram [63], systematic search B; Figure S3: Study characteristics, search A; Figure S4: Study characteristics, search B; Table S21: Results of quality appraisal with CASP checklists, search A and B; Table S22: Original copy of CONSIDER checklist from Huria et al. [76]; Table S23: CONSIDER checklist scored for two included articles [41,79] of search B; Table S24: Weight, or ranking, given to the protective factors, search B.

**Author Contributions:** Conceptualization, V.S.M.V.; methodology, V.S.M.V.; formal analysis, V.S.M.V.; investigation, V.S.M.V.; data curation, V.S.M.V.; writing—original draft preparation, V.S.M.V.; writing—review and editing, V.S.M.V., C.S. and P.M.; supervision, C.S. All authors have read and agreed to the published version of the manuscript.

**Funding:** This research received no external funding.

**Data Availability Statement:** All data supporting results will be made publicly available in an Supplementary File at: www.mdpi.com/xxx/s1 (see above: Supplementary Materials and further information can be requested by sending an e-mail to the first author (V.V.)).

**Acknowledgments:** The first author thanks the municipal public health service (GGD), department of environment and health, located in Groningen, for their support.

**Conflicts of Interest:** The authors declare no conflicts of interest.

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
