# Peer review of "Voices from the North: Exploring Sámi People’s Perspectives on Environmental Change and Mental Well-Being: A Systematic Literature Review†"

_challenges, doi:10.3390/challe15020030_

Round 1

Reviewer 1 Report

Comments and Suggestions for Authors

It is extremely important to be aware of the impact of climate change on unique cultures. This paper will therefore make a unique contribution by summarizing the literature that is currently out there.

The authors are very good writers and I did not find spelling or grammar issues which is wonderful! However, as a reader I found the structure of the paper very confusing. I almost feel that it should be two papers- one focusing on circumpolar Indigenous communities and one focusing on Saami communities. The back and forth structure was confusing. You could also split the two findings apart summarizing all of the literature for the circum- polar Peoples in section 1. Section 2 could be summarizing the Saami literature. Section 3 could be a comparison. Section 4 model of protective factors and resilience (see request fro you to define these). Recommend you delete Figure 3 as it is confusing or place it in section 3.

I also would like to see an acknowledgement of the authors' non-Indigenous identities up front and the recognition that all literature is therefore incomplete and potentially biased. By acknowledging this, you will increase your credibility.

In summary, the topic is very important. The writing and thinking is sound. It is only the structure and organization that need work. Address these issues and the ones in the review and this paper will offer an important contribution to the literature. 

I hope you find this helpful.

Reviewer 2 Report

Comments and Suggestions for Authors

Thank you for the opportunity to read your excellently accomplished, two-pronged systematic review of literature pertaining to the broader Circumpolar Indigenous community and to the Sámi People as well as an insightful thematic comparison of findings from both. It appears that your findings could be applied for use in both policy initiatives as well as for the development of targeted interventions. And they could also be used by researchers seeking to follow through with investigations of strength-based factors such as appear to be operational in the holistic perspective adopted by Circumpolar (Sámi) Indigenous Peoples. Incidentally, one of the great benefits of reviewing current work is coming across literature as-yet unbeknownst to me, and I have some new takeaways thanks to your review. 

I noted some discrepancies in the review and have listed these as follows: 

L 62 ‘strength-based’ approach, but elsewhere ‘strength based’ (e.g., l 66, 76)

L 78. Need ending single quotation mark: ‘factors promoting resilience… 

L 228. “table 2”; title case for Table

L 231. Redundant full stop at end of Table 2 title. 

L 283. Need ending single quotation mark or remove first mark: ‘encounters with environmental changes…

L 372 “are presented table 3”…. ‘are presented in Table 3’

L 398. Need ending single quotation mark or remove first mark: “‘encounters with environmental…

L 411. Clarify what is meant by ‘the encroachment of the treeline’. Is that something is encroaching onto the treeline? Or that the treeline is encroaching onto something else? 

L 421-423. Just an observation as I was reading; this passage reads like a tragedy of the commons re Hardin. 

“This intensified grazing pressure, coupled with diminishing grazing lands, perpetuates a cycle of ecological degradation and financial strain, compelling herders to further increase herd sizes, thereby exacerbating the unsustainable trajectory [75,77].”

L 535. “All factors, except for ‘physical exercise’…”. ‘Physical activity’ in Figure 1. 

L 642. “These are summarised in figure 2.” Title case for Figure

L 654. Need ending single quotation mark: ‘journaling, solitary retreats…

L 663. “Figure 7 encapsulates…” Figure 3. 

L 668. “figure 7” Figure 3. 

L 766. “strength-based”. Everywhere else you’ve used single quotation marks. 

L 778. Delete hanging apostrophe: Focusing on perspectives’,

Round 2

Reviewer 1 Report

Comments and Suggestions for Authors

The authors have addressed all of my concerns and added content to make the paper more clear. They have reformatted the content so that it flows well and allows the observations to stand out. I also appreciated their upfront acknowledgement of their non-Indigenous identity.

Nicely done!